# Maternal and Neonatal Outcome after the Use of G-CSF for Cancer Treatment during Pregnancy

**DOI:** 10.3390/cancers13061214

**Published:** 2021-03-10

**Authors:** Claudia Berends, Charlotte Maggen, Christianne A. R. Lok, Mathilde van Gerwen, Ingrid A. Boere, Vera E. R. A. Wolters, Kristel Van Calsteren, Heidi Segers, Marry M. van den Heuvel-Eibrink, Rebecca C. Painter, Mina Mhallem Gziri, Frédéric Amant

**Affiliations:** 1Center for Gynecological Oncology Amsterdam, Antoni van Leeuwenhoek—Netherlands Cancer Institute, Plesmanlaan 121, 1066 CX Amsterdam, The Netherlands; clberends@rijnstate.nl (C.B.); c.lok@nki.nl (C.A.R.L.); m.v.gerwen@nki.nl (M.v.G.); v.wolters@nki.nl (V.E.R.A.W.); 2Department of Oncology, KU Leuven, 3000 Leuven, Belgium; charlotte.maggen@kuleuven.be; 3Department of Obstetrics and Prenatal Medicine, Vrije Universiteit Brussel (VUB), University Hospital of Brussels, 1090 Brussels, Belgium; 4Princess Máxima Center for Pediatric Oncology, 3584 CS Utrecht, The Netherlands; m.m.vandenheuvel-eibrink@prinsesmaximacentrum.nl; 5Department of Medical Oncology, Erasmus MC Cancer Institute, Erasmus University Medical Center, 3015 CN Rotterdam, The Netherlands; i.boere@erasmusmc.nl; 6Department of Obstetrics and Gynecology, University Hospitals Leuven, 3000 Leuven, Belgium; kristel.vancalsteren@uzleuven.be; 7Department of Development and Regeneration—Unit Woman and Child, KU Leuven, 3000 Leuven, Belgium; 8Department of Pediatric Hemato-Oncology, UZ Leuven, 3000 Leuven, Belgium; heidi.segers@uzleuven.be; 9Department of Obstetrics and Gynecology, Amsterdam University Medical Centers, Amsterdam Reproduction and Development, University of Amsterdam, 1105 AZ Amsterdam, The Netherlands; r.c.painter@amsterdamumc.nl; 10Department of Obstetrics, Cliniques Universitaires St Luc, UCL, 1200 Sint-Lambrechts-Woluwe, Belgium; mina.mhallem@uclouvain.be; 11Center for Gynecological Oncology Amsterdam, Amsterdam University Medical Centers, 1105 AZ Amsterdam, The Netherlands

**Keywords:** pregnancy, cancer treatment, G-CSF, maternal outcome, neonatal outcome

## Abstract

**Simple Summary:**

Treatment of pregnant cancer patients should adhere as much as possible to standard treatment protocols in order to safeguard maternal prognosis. The use of Granulocyte colony-stimulating factor (G-CSF) can be indicated for dose dense chemotherapy in high risk breast cancer patients or for the treatment or prevention of neutropenic fever, which can be an important threat for maternal and fetal survival. However, as evidence is still scarce, physicians are still reluctant to the use of G-CSF during pregnancy. In this series, the International Network of Cancer, Infertility and Pregnancy (INCIP) reports on 42 pregnant patients who received G-CSF during oncological treatment. Reported maternal and neonatal complications are acceptable; however, a continuous evaluation of clinical practice is necessary as the limited data in numbers and follow-up do not allow robust conclusions.

**Abstract:**

Data on the use of Granulocyte colony-stimulating factor (G-CSF) in pregnant cancer patients are scarce. The International Network of Cancer, Infertility and Pregnancy (INCIP) reviewed data of pregnant patients treated with chemotherapy and G-CSF, and their offspring. Among 2083 registered patients, 42 pregnant patients received G-CSF for the following indications: recent chemotherapy induced febrile neutropenia (5; 12%), dose dense chemotherapy (28, 67%), poly chemotherapy (7, 17%), or prevention of neutropenia at delivery (2; 5%). Among 24 women receiving dose dense chemotherapy, three (13%) patients recovered from asymptomatic neutropenia within 5 days. One patient developed pancytopenia following polychemotherapy after which the pregnancy was complicated by chorioamnionitis and intrauterine death. Nineteen singleton livebirths (49%) were born preterm. Sixteen neonates (41%) were admitted to the Neonatal Intensive care Unit (NICU). No neonatal neutropenia occurred. Two neonates had congenital malformations. Out of 21 children in follow-up, there were four children with a motor development delay and two premature infants had a delay in cognitive development. In conclusion, the rate of maternal and neonatal complications are similar to those described in (pregnant) women treated with chemotherapy. Due to small numbers and limited follow-up, rare or delayed effects among offspring exposed to G-CSF in utero cannot be ruled out yet.

## 1. Introduction

The co-occurrence of cancer and pregnancy, estimated to affect 1 in 1000 pregnancies, is expected to rise due to increasing maternal age and incidental findings at the occasion of the non-invasive prenatal testing (NIPT) [1,2]. Breast cancer, melanoma, hematological, and cervical cancer are the most common types of cancer diagnosed during pregnancy [1,3]. With the increasing awareness of the feasibility of antenatal cancer treatment, fewer pregnancies are terminated and more pregnant women receive chemotherapy [4]. Whenever possible, oncological treatment during pregnancy should adhere as much as possible to the standard of care treatment in non-pregnant patients, in order to safeguard prognosis [5]. However, consequences and safety of some supportive agents that are used as part of current standard therapy, such as Granulocyte Colony-Stimulating Factor (G-CSF), are still subject of discussion.

G-CSF supports the clonal growth of progenitors of neutrophils and regulates the proliferation and differentiation of hematopoietic stem cells. Both in vitro and in vivo studies confirmed transplacental passage of this glycoprotein [6,7]. In oncological care, it is used to treat or prevent prolonged grade 3 (absolute neutrophil count (ANC) < 1.0 × 10^9^/L) and 4 (ANC < 0.5 × 10^9^/L) neutropenia or febrile neutropenia (grade 3–4 neutropenia with fever) in patients receiving chemotherapy [8,9]. In patients with high risk breast cancer, dose dense chemotherapy regimens with G-CSF support, are considered standard of care [5,10].

As available data are still limited to case reports and small case series, use of antenatal G-CSF is still debated [11,12]. Hence, more maternal efficacy and neonatal safety data in larger cohorts are necessary. The aim of this cohort study is to describe the clinical outcomes after use of G-CSF in pregnancy, as part of cancer treatment, in patients and their offspring registered by the International Network of Cancer, Infertility and Pregnancy (INCIP).

## 2. Materials and Methods

Women with cancer and treated with G-CSF during pregnancy were identified in the INCIP database. The INCIP study has been approved by the ethical committee of the university hospitals of Leuven in Belgium (S25470) and the Erasmus Medical Center in the Netherlands (NL4354607813). The international multicenter study is registered on ClinicalTrials.gov, number NCT00330447. The registry contains both retro- as prospectively collected obstetric and oncological data of women (and their offspring) with a cancer diagnosis in association with pregnancy.

The INCIP database was reviewed for oncological, obstetrical, neonatal, and pediatric data, and missing data were requested from participating hospitals. Pregnancy dating was confirmed in all patients by early ultrasound. To define the efficacy of G-CSF in prevention of chemotherapy-induced neutropenia (dose dense regimen or polychemotherapy regimens with a high risk of neutropenia), results of all available maternal blood samples taken before and during chemotherapy until two weeks after the last administration of G-CSF were retrospectively extracted from patient files. The incidences of maternal neutropenia, leukopenia, anemia, and thrombocytopenia were assessed. Grading of neutropenia, leukopenia, anemia, and thrombocytopenia was defined according to the Common Terminology Criteria for Adverse Events (CTCAE version 5.0) [8]. Neutropenia was divided in grade 1–2 (mild) (absolute neutrophil count (ANC) 1.5 × 10^9^/L—lower limit of normal (LLN) or ANC 1.0–1.5 × 10^9^/L, respectively) and grade 3–4 neutropenia (ANC of 1.0–0.5 × 10^9^/L or below 0.5 × 10^9^/L, respectively). Febrile neutropenia was defined as an ANC below 1.0 × 10^9^/L and fever (=>38 °C). Grade 3–4 leukopenia was defined as a white blood cell count (WBC) below 1.0 × 10^9^/L), thrombocytopenia was divided in grade 1–2 (mild) [platelets count (PC) < 150 × 10^9^/L) and grade 3–4 (severe) (PC < 50 × 10^9^/L) [8]. Anemia was defined as grade 1–2 (mild) (hemoglobin (Hb) 8–10 g/dL) and 3–4 (severe) anemia (Hb level below 8.0 g/dL). Oncological data that were collected were tumor type, chemotherapy regimen, gestational age (GA) at the start of chemotherapy and number of G-CSF administrations.

The neonatal blood samples were taken within 48 h after delivery. Neonatal neutropenia was defined as absolute neutrophil count (ANC) < 1.0 × 10^9^/L, leukopenia as white blood cell count (WBC) < 5.0 × 10^9^/L, thrombocytopenia as PC < 15 × 10^9^/L, and anemia as Hb less than 14 g/dL [8,13,14,15,16,17]. Customized percentiles for birthweight (p) were calculated, adjusted for GA at delivery, parity, ethnicity, body mass index (BMI), and sex of the infant [18]. Neonates were small for gestational age (SGA) if the birthweight was below the 10th percentile. Neonatal outcomes of two twin pregnancies were described separately.

In addition, available data of children included in the long-term prospective follow-up study of the INCIP project were collected. In this follow-up study, children underwent a general physical examination by a pediatrician (including clinical neurological evaluation), cognitive (neuropsychological) tests by a psychologist (see Appendix A for details), and cardiac evaluation (electrocardiogram (ECG) and echocardiography) by a cardiologist at different time-points. To assess maternal and neonatal outcomes, descriptive analysis (percentages, median and range) were performed.

## 3. Results

Out of 2083 patients registered by INCIP, 42 patients with cancer during pregnancy were treated with chemotherapy and G-CSF (Figure 1). The majority of patients was diagnosed with breast cancer (*n* = 35, 83%), followed by non-Hodgkin lymphoma (*n* = 5, 5%), Ewing sarcoma (*n* = 1, 1%), and acute lymphocytic leukemia (ALL) (*n* = 1, 1%) (Table 1).

### 3.1. Treatment

The median GA at the start of chemotherapy was 22 weeks (range 11–36). One patient started chemotherapy for stage 3 breast cancer at 11 weeks of gestation and delivered of twins without malformations, in all other patients chemotherapy was initiated after 13 weeks of gestation. Both short-acting (filgrastim) and long-acting G-CSF (pegfilgrastim, lipegfilgrastim) were administered for the following indications:-Long-acting G-CSF was mostly given as part of a dose dense schedule (*n* = 28; 67%).-Five patients (12%) developed grade 3–4 neutropenia (including one patient with neutropenic fever) after one to three cycles of 3-weekly chemotherapy (without G-CSF). Two of them received filgastrim during the acute episode of neutropenia and all five patients had long-acting G-CSF with the subsequent chemotherapy (without treatment delay) administrations in prevention of febrile neutropenia or dose delays.-Seven women (7%) received long-acting G-CSF following ‘high risk’ polychemotherapy for Non Hodgkin lymphoma (*n* = 4), Ewing sarcoma (*n* = 1) or ALL (*n* = 1).-Long-acting G-CSF (pegfilgastrim) was given prophylactically after the last chemotherapy before delivery in two women (5%).

### 3.2. Maternal Blood Results

Blood results of 24 women who received dose dense chemotherapy were registered (Table 2). Uncomplicated neutropenia grade 3–4 occurred in three women (13%) with breast cancer, but all recovered within five days. There were no reports of febrile neutropenia, nor thrombocytopenia. Mild anemia occurred in 14 women (58%) and severe anemia in two (8%) women.

Three patients with available blood counts received long-acting G-CSF as part of intense polychemotherapy with a high risk for neutropenia. One patient suffered from a pancytopenia severe anemia, severe thrombocytopenia, and WBC < 0.1 × 10^9^/L) following vincristine, ifosfamide, doxorubicin, etoposide (VIDE) chemotherapy. Two patients received rituximab, cyclophosphamide, doxorubicin, vincristine, prednisone (R-CHOP) with pegfilgastrim and did not develop hematological toxicities, except for mild anemia in one patient.

### 3.3. Obstetric Outcome

The obstetric outcomes of 40 singleton pregnancies are described in Table 3. One patient developed febrile neutropenia and secondary pharyngitis following the first course of 5-fluorouracil, epirubicin, cyclophosphamide (FEC) chemotherapy (without G-CSF). Three other patients had a maternal infection requiring antibiotics (two patients with pneumonia, one with a postpartum systemic infection) without leukopenia/neutropenia. An earlier mentioned patient developed a pancytopenia and chorio-amnionitis (without any prior invasive prenatal procedures) and spontaneously delivered a growth-restricted stillborn neonate at a GA of 23 weeks. All other 41pregnancies ended in live births (including two twin pregnancies). Labor was induced in 18 patients (45%), mostly because of maternal therapy planning (78%). Of the 19 pregnancies that ended pre-term (49%), 12 had a planned delivery, of whom 8 (67%) for therapy planning. The seven other women delivered preterm after spontaneous onset of labor. In total, 23 women (55%) delivered within 3 weeks after the last chemotherapy administration, of whom 11 women (25%) delivered within two weeks (7 spontaneous labors, two emerging obstetrical reasons, two planned deliveries for oncological treatment). Emergency caesarean sections were performed in three women because of pre-eclampsia, fetal distress, and prolonged second stage of labor, respectively.

### 3.4. Neonatal Outcome

Nineteen singleton livebirths were born preterm (19 of 39, 49%). Of all singleton neonates, 11 (28%) were SGA. Thirteen neonates (41%) were admitted to the NICU, mainly because of prematurity (81%) (Table 4). There were no reports of neonates with leukopenia or neutropenia and there were two neonates, born term, with anemia (Hb measurements between 12.0 and 13.5 g/dL). One neonate born one day after prenatal polychemotherapy by emergency cesarean section at a GA of 29 weeks for fetal distress, suffered from a Klebsiella sepsis with thrombocytopenia and intravascular coagulation leading to microthrombi. Two other neonates, delivered preterm within 3.5 weeks following chemotherapy, were treated with antibiotics because of systemic infection. Two neonates had a congenital malformation: one neonate prenatally exposed to 5-FU, epirubicin, cyclophosphamide (FEC) (+ pegfilgastrim) and docetaxel from 20 weeks of pregnancy onwards, was born with an absent uvula. The other neonate, prenatally exposed to dose dense doxorubicin, cyclophosphamide (AC) between 30 and 36 weeks of gestation, had a severe pulmonary valve stenosis, diagnosed after birth. It was successfully treated with balloon dilatation at the age of 1 month. In both cases, there were no maternal risk factors for congenital malformations, nor neonatal chromosomal abnormalities reported.

One women delivered of healthy twins following induced labor at 37 weeks after dose dense epirubicin, cyclophosphamide (EC) and paclitaxel weekly. Another twin pregnancy ended in a spontaneous delivery at 29 weeks after dose dense EC and paclitaxel weekly.

### 3.5. Pediatric Outcome

Twenty-one children participated in the follow-up study of the INCIP. The median follow-up was 18 months (range 2 months–9 years) (Appendix A). No neurological or functional cardiac abnormalities were observed. In four children (19%), born at a median GA of 38 weeks, motor development was delayed at 11 months (*n* = 1) and 18 months (*n* = 3). One child had a hip dysplasia and one child had a preferential posture, which both required physiotherapy. Eventually, all four children had a normal motor development at 18 months and 36 months of follow-up according to standardized and clinical measures of development. In four children (19%), delayed cognitive development was identified; one child born at a GA of 29 weeks and 2 days, was assessed at 18 months of age and had an appropriate cognitive development, but delayed language development. The child was referred to a speech-language therapist. Two children (10%), both born after GA of 37 weeks, had a delay in cognitive development at 18 months of age, but cognitive development was appropriate at 3 and 6 years of age according to standardized and clinical measures of neurocognitive development. Another child, prenatally exposed to nicotine, cannabis and chemotherapy, was born at a GA of 35 weeks and 4 days with a birthweight of 2060 g (P 5.1). The child had an appropriate cognitive development at 18 months of follow-up, but cognitive and language development was delayed at 3 years of age. At the age of 3 years, the child also had behavioral and emotional problems and bodyweight was above the 97th percentile. The child and parents were referred to a specialized center for childcare.

## 4. Discussion

In this manuscript, we report the maternal and neonatal outcomes of 42 patients with chemotherapy and G-CSF treatment during pregnancy. In the 24 patients who received dose dense chemotherapy supported with G-CSF, febrile neutropenia did not occur and grade 3–4 maternal neutropenia could often be prevented (*n* = 3; 13%). There was one stillbirth following maternal pancytopenia and chorio-amnionitis after VIDE chemotherapy, despite administration of G-CSF to prevent hematological toxicity. Two out of 39 singletons had a congenital malformation (5%). Neonatal neutropenia did not occur and there were no major abnormalities reported in the clinical follow-up of 21 children.

Of note, G-CSF in pregnancy should only be considered when there is a clear indication. There is no evidence to support prophylactic use of the drug prior to delivery, as done in two cases in this series. Outside pregnancy, the addition of G-CSF to chemotherapy improves overall survival, as it minimizes treatment delays and allows dose-dense chemotherapy regimens leading to an increased disease control [19]. The risk of neutropenic fever during chemotherapy in pregnancy is unknown, but may be lower than outside pregnancy owing to the more rapid clearance of chemotherapy and larger distribution volume in pregnancy [20]. However, the consequences of febrile neutropenia in pregnancy threaten both maternal and fetal survival. G-CSF is usually well tolerated, with medullary bone pain being the most frequently reported side effect [21]. Other less common adverse effects include headaches, generalized musculoskeletal pain and, very rare, an anaphylactic-like reaction. An increased risk of secondary hematological malignancies in cancer patients receiving G-CSF is suggested, although this association has not been found consistently and might be also related to increased doses of chemotherapeutic agents with leukemogenic potential [19,22]. Moreover, patients with severe congenital neutropenia treated with G-CSF, are at long-term risk to develop myelodysplastic syndrome and acute myeloid leukemia [23]. Another concern is that G-CSF might contribute to a hypercoagulable state and thrombosis, besides other factors in this high risk population (pregnancy, cancer, surgery, and chemotherapy) [24,25].

Physicians are hesitant to use antenatal G-CSF as it crosses the placenta and could affect the development of the unborn child, including spontaneous miscarriage and congenital malformations [26]. Although there are reassuring data on G-CSF use in neonates, these data cannot just be generalized to the fetus because of the immature fetal metabolism and organs [27]. The neonatal consequences of G-CSF in pregnancy have mainly been investigated for treatment of neutropenia unrelated to chemotherapy. Four large studies and five case reports, with in total 162 pregnancies, have investigated G-CSF in pregnancy for treatment of chronic neutropenia [26,28,29,30,31,32,33,34,35]. In these studies, G-CSF administration ranged from the first to the third trimester. None of these studies found an increased incidence of fetal death or congenital malformations. In four pregnancies where G-CSF was used because of ritodrine (a tocolytic drug)-induced neutropenia, no maternal or neonatal adverse effects of G-CSF were found [36,37]. Furthermore, it is suggested that G-CSF can be administered in pregnancy or lactation in order to mobilize stem cells for stem cell transplantation [38]. Although these data are reassuring concerning G-CSF use, patients were not comparable to our cohort as they did not receive (dose dense) chemotherapy, which is an extra risk factor for adverse maternal and neonatal outcome.

Neonates born from 12 mothers who received G-CSF just before delivery were shown to have increased neutrophil counts compared to a control group [7]. La Nasa et al. reported an incidence of neonatal neutropenia and leukopenia after chemotherapy and long-acting G-CSF of only 4% (*n* = 24 and *n* = 26, respectively) [15]. Using the same definitions, we did not observe any neonatal neutropenia or leukopenia in this series. These results suggest that G-CSF may not only be beneficial for the mother, but also for the neonate as the prevention of leucopenia will reduce the risk of infection. In this series, there were three neonates with an early onset systemic infection without maternal neutropenia. All three neonates were born preterm (between GA of 28–34 weeks) and were SGA, both risk factors for neonatal sepsis [39].

The incidence of SGA in this population was 28% and 41% of neonates were admitted to the NICU, mostly because of prematurity, which is comparable to large cohort studies on pregnant women with cancer (21% and 41%, respectively) [4]. Major congenital malformations are estimated to occur in 255 out of 10,000 births (2.5%) [40]. In this series, two birth defects (5%) were reported: pulmonary valve stenosis and absent uvula. The latter is an extremely rare condition and based on the registered data it is not clear whether this malformation was isolated or part of a congenital syndrome. Of note, G-CSF and chemotherapy were administered in all patients after the most vulnerable period of fetal organogenesis (between 2 and 8 weeks following conception) [41]. As G-CSF has no cytotoxic mechanism of action, it is assumable that the causality with congenital malformations is unlikely, but safety in the first trimester cannot be guaranteed based on this series. In comparison, the occurrence of congenital malformation in the pregnant cancer population is reported to be 4% (2% major and 2% minor malformations according to EUROCAT) [4,42]. In total, 22 (3.0%) major and 13 (1.8%) minor congenital malformations were seen in the offspring of 726 women treated with chemotherapy in the second and third trimester of pregnancy (INCIP data not published).

The rate of severe neutropenia following dose dense chemotherapy was 13% (3/24), which is comparable with the reported rates in the non-pregnant population (14.9%) [43]. However, the decrease in leukocytes following chemotherapy in pregnancy might be underestimated when reference counts from the general population are used, as during pregnancy the leukocyte count significantly increases physiologically [44]. The observed rate of severe anemia was higher compared to the non-pregnant population (8% vs. 2%), however the incidence of mild anemia was comparable (58% vs. 66%) [43]. Of note, gestational changes induce a ‘physiological dilutional’ anemia, resulting in reduced physiological Hb levels (10 to 11 g/dL), but severe anemia is unlikely to be explained by pregnancy alone [45]. Of note, there was no report of febrile neutropenia nor thrombocytopenia following dose dense treatment in this series.

Major neurologic or functional cardiac abnormalities were not found during follow-up of 21 children. Earlier studies showed that prematurity is a predictor for worse cognitive outcome rather than prenatal exposure to cancer treatment [46]. In addition to this series, no significant difference in incidence of behavioral problems, asthma, eczema, or problems with speech were found between 29 exposed children, with a mean follow-up of 54 months, and 114 non-exposed children [12]. Of note, available data are still too limited to make robust conclusions and highlights the need for continuous follow-up, especially as children born SGA seems to be at risk for neurological dysfunction [47].

The maternal and neonatal outcomes in this series are in line with previously published cohort studies of 10 and 34 pregnant women that received G-CSF during oncological treatment [11,12]. A strength of this study was the information on pediatric outcomes after antenatal G-CSF, collected as part of an international registration study. The retrospective nature of the registration study incorporates inevitably missing data. Reported numbers are too low to distinguish consequences of short- and long-acting G-CSF separately. The follow-up of children in this series (maximum 9 years) is also too short to learn about childhood malignancies. Research in larger cohorts remains indispensable to confirm the independent benefit and low incidence of adverse events of G-CSF in the pregnant population and their offspring.

## 5. Conclusions

Since we did not observe any marked increase in perinatal complications and the outcomes of this series are in line with available literature, we conclude with caution that the use of G-CSF during pregnancy can be considered when this is clinically indicated for maternal oncological treatment. However, our study was not powered for perinatal complications with low incidence, such as thrombosis, or delayed long-term effects, including secondary malignancies. G-CSF should therefore continue to be administered with caution, and only for indications which provide proven benefit for survival and cancer prognosis, and preferably in the context of ongoing registration studies. These data further contribute to the policy to treat pregnant cancer patients as much as possible as non-pregnant cancer patients in order to safeguard cancer outcomes.

## Figures and Tables

**Figure 1 cancers-13-01214-f001:**
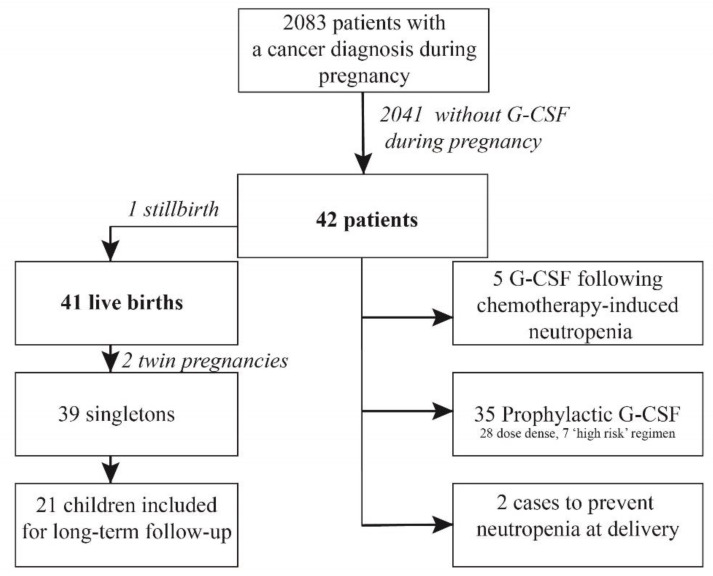
Study Flow Chart.

**Table 1 cancers-13-01214-t001:** Maternal characteristics (*n* = 42).

Maternal Characteristics	Median	Range
Age at diagnosis (years)	34	19–47
BMI at booking (kg/m^2^) *	25.9	18.3–36.9
	**Number**	**%**
**Ethnicity**		
Caucasian	32	76.2
Non-caucasian	6	14.3
Not reported	4	9.5
**Type of malignancy**		
Breast cancer	35	83.3
Non Hodgkin lymphoma	5	11.9
Ewing sarcoma	1	2.4
Acute lymphocytic leukemia	1	2.4
**Treatment modality**		
Chemotherapy	28	66.7
Chemotherapy + surgery	14	33.3
**Chemotherapy**		
Anthracycline-based	18	42.9
Anthracycline-based with taxanes	15	35.7
Other **	9	21.4
	**Median**	**Range**
Gestational age at first chemo (weeks)	22	11–36
Cycles of chemotherapy during pregnancy	6	1–16
Administrations of G-CSF	4	1–16
	**Number**	**%**
**Indication G-CSF**		
Prophylactic in dose dense chemotherapy	28	66.7
Propyhlactic in polychemotherapy regimen	7	16.7
Prophylactic before delivery	2	4.8
Following chemotherapy-induced neutropenia	5	11.9
**Type of G-CSF**		
Pegfilgrastim	28	66.7
Lipefilgrastim	8	19.1
Filgrastim	3	7.1
Not reported	3	7.1

* for 5 patients, BMI was not available. ** Other chemotherapy regimens include rituximab, doxorubicin, cyclophosphamide, vincristine, prednisone (R-CHOP), etoposide, cisplatin (EP), vincristine, ifosfamide, doxorubicin, etoposide (VIDE), rituximab, doxorubicin, cyclophosphamide, vindesine, bleomycin, prednisone (R-ACVPB), epirubicin, cyclophosphamide, paclitaxel, carboplatinum (EC+TC), ifosfamide, etoposide, without methotrexate during pregnancy (VIM), and hyper-CVAD course A (cyclophosphamide, vincristine, doxorubicin, cytarabine, no Methotrexate during pregnancy).

**Table 2 cancers-13-01214-t002:** Maternal blood results following dose dense chemotherapy during pregnancy (*n* = 24 with available serial blood tests following G-CSF).

Maternal Blood Results	Total *n* *		Grade 1–2		Grade 3–4	
	*n*	%	*n*	%	*n*	%
Neutropenia	5	21	2	8	3 *	13
Leukopenia *	4	17	2	8	2	8
Thrombocytopenia	0	0	0	0	0	0
Anemia	16	67	14	58	2	8

* one patient had grade 3 neutropenia (0.58 × 10^9^/L) without leukopenia (3.2 × 10^9^/L).

**Table 3 cancers-13-01214-t003:** Obstetric outcomes (*n* = 40, all singleton pregnancies).

Obstetric Outcomes	*n*	%
Live birth	39	97.5
Still Birth	1	2.5
**Gestational age at delivery** (*n* = 39 *)		
≥37 weeks (a term)	20	51.3
<37 weeks (pre term)	19	48.7
**Onset of labor**		
Spontaneous	13	32.05
Induction of labor	18	45.0
Cesarean section	9	22.5
Emergency (fetal distress)	2	22.2
Elective (all for obstetrical reason **)	7	77.8
**Reason induction of labor (*n* = 18)**		
Obstetrical reason ***	1	5.6
Therapy planning	14	77.8
Deterioration of maternal condition	2	11.1
Other	1	5.6
**Mode of delivery**		
Vaginal delivery	27	67.5
Assisted vaginal delivery	2	5.0
Elective Cesarean section	8	20.0
Emergency cesarean section	3	7.5
**Complications**		
Maternal infection (including 1 chorioamnionitis)	4 (1 postpartum)	10.0
Gestational diabetes	1	2.5
Preeclampsia	1	2.5
Maternal neutropenia/leukopenia	7	17.5
PPROM or preterm contractions	9	22.5
Stillbirth	1	2.5
Postpartum hemorrhage	2	5.0

* 1 stillbirth was excluded. ** placenta previa, repeat cesarean section, breech. *** hypertension, preeclampsia, cholestasis, diabetes, premature rupture of membranes.

**Table 4 cancers-13-01214-t004:** Neonatal outcomes (*n* = 39, all singleton live births).

Neonatal Outcomes	Median	Range
Birth weight (grams)	2855	850–3780
	***n***	**%**
**Customized birthweight percentile**		
<10	11	28.2
11–49	18	46.1
50–89	8	20.5
>90	2	5.1
**APGAR at 5 min**		
4	1	2.6
9	8	20.5
10	29	74.4
Not reported	1	2.6
**Congenital malformation**		
yes	2	5.1
no	37	94.8
**Admission neonatal care unit**		
yes	16	41.0
no	20	51.3
Not reported	3	7.7
**Reason admission neonatal care unit**		
Prematurity	13	81.3
Observation because of maternal chemotherapy	1	6.3
Other	2	12.5
**Neonatal blood results**		
Leukopenia *	0	0
Neutropenia **	0	0
Thrombocytopenia ***	1	5.8
Anemia ****	2	10
**Neonatal complications**		
Hyperbilirubinemia	2	5.2
Neonatal sepsis	3	7.7
First degree cerebral bleeding related to prematurity	1	2.6

* Available WBC *n* = 24, ** Available ANC *n* = 12, *** Available PC *n* = 18; **** Available hemoglobin *n* = 20.

## Data Availability

The data presented in this study are available upon request from the corresponding author. The data are not publicly available.

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
