# Peer review of "Maternal and Neonatal Outcome after the Use of G-CSF for Cancer Treatment during Pregnancy"

_cancers, 2021, doi:10.3390/cancers13061214_

Round 1

Reviewer 1 Report

In this series the authors reported the maternal and fetal outcome of 42 pregnant patients that received G-CSF during 34 oncological treatment within the the International Network of Cancer, 33 Infertility and Pregnancy (INCIP) reports.

The paper is very well written, exhaustive, and really interesting. I suggest the following revisions to be suitable for publication:

  • there are some typo in the whole text, and missing punctuation should be corrected
  • reference 9 should be updated with the most recent version 10.1200/JCO.2015.62.3488 Journal of Clinical Oncology 33, no. 28 (October 01, 2015) 3199-3212.
  • reference 10 should be replaced with the individual patient meta-analysis conducted by EBCTCG (10.1016/S0140-6736(18)33137-4)
  • the figure 1 is cut at the bottom, please modify it
  • If authors had data about the incidence of typical toxicities of G-CSF (for example, bone pain, bone pain requiring use of analgesics), they should be reported in the results section

Reviewer 2 Report

The article by Berends et al., seeks to evaluate the maternal and neonatal outcome after the use of GCSF during pregnancy. These data are important and are needed to understand the possibilities of side effects due to use of the treatment as well as sharing how they could be safe. However, this paper is a simple report, and not an article paper that are at a level of publication. This reviewer understands that it is hard to obtain numbers of patients who went through these treatments but even with small numbers, it is crucial to compare to the normal range. Authors neglected to compare any of their data to the normal pregnancies or any other treatments that are used during pregnancies. At this point, there are no conclusion where there are possibilities that GCSF could be a useful treatment during pregnancy or it should be considered harmful.

Reviewer 3 Report

In this study, Berends et al. reported the maternal and neonatal outcomes of a cohort of 42 patients with chemotherapy and Granulocyte colony-stimulating factor (G-CSF) treatment during pregnancy. In their study, the authors revealed that among the cohort 24 patients that received dose dense chemotherapy supported with G-CSF, febrile neutropenia did not occur, and grade 3-4 maternal neutropenia could often be prevented. Moreover, they revealed that there is only one stillbirth following maternal pancytopenia and chorio-amnionitis after VIDE chemotherapy, despite administration of G-CSF to prevent hematological toxicity. On the other hand, they showed that only two out of 39 singletons had a congenital malformation (5%). Moreover, they did not observe any major abnormalities in the clinical follow-up of 21 children for 9 years which can be considered as a short period to learn about this kind of management and the possibility of developing different types of diseases including cancer. Thus, this is a quit interesting paper; however, and as the authors mentioned in the discussion: “Four large studies and five case reports, with in total 162 pregnancies, have investigated G-CSF in pregnancy for treatment of chronic neutropenia. [25, 27-34] In these studies G-CSF administration ranged from the first to the third trimester. None of these studies found an increased incidence of fetal death or congenital malformations.” Thus, I cannot see what this study can add to the literature and especially if we can add the limited number of cases studied and the relatively short period of follow-up.

Reviewer 4 Report

Congratulations to the authors. They provide interesting data about C-GSF use during pregnancy. This is a really interesting manuscript that helps us to be confident about the G-CSF use during pregnancy. It is well writing and explains well the treatment and evolution of those patients and I just will ask for minor corrections in order to do it more comprehensible.

Introduction: Nothing to comment

Material and Methods:

According to table 1, the range of gestational age at 1st chemo was 11. This is not a standard procedure as the first trimester is not ended. Although more sensitive organogenesis is ended there can be some concerns as posterior palate formation is not ended (Moore Pernsaud The developing Human:  Clinically Oriented Embryology, WB Saunders Philadelphia). A brief explanation of how many cases were treated before the end of the first trimester and why is needed.

Gestational age is a key point for those patients. Please confirm that all patients had a first-trimester sonography assessment and that GA is corrected according to this sonography.

One neonate had pulmonary valve stenosis. Please provide some data of when chemotherapy is started, the grade of stenosis, and if it could be diagnosed prenatally by focused cardiac ultrasound.

There is one case of chorioamnionitis. Please specify that there were not invasive procedures before it that can explain the disease.

Discussion: Nothing to comment

Conclusions: Well exposed and related to the observed results

Reviewer 5 Report

There are very small typographical mistakes:

line 31:  "treat" should be "threat"

Tables 3 and 4: the title of the page is on separate page

line 247 "we did not even observe one neonate" could be "we did not observe any neonate" (or something similar)

line 291 "because" should be "since" (or other similar adverbs)

Round 2

Reviewer 2 Report

Although this reviewer still thinks that this study is just a simple set of report and is not a quality of an article to give any conclusions, it is understandable that sample improvements are hard. This reviewer would like to leave the final decision to the editor.

Reviewer 3 Report

Authors did not address my comments which is: "I cannot see what this study can add to the literature and especially if we can add the limited number of cases studied and the relatively short period of follow-up.